# Automatic Pavement Crack Detection Fusing Attention Mechanism

**Junhua Ren [1], Guowu Zhao [1], Yadong Ma [2], De Zhao [3], Tao Liu [1,*] and Jun Yan [2]**

[1]  Linyi Highway Development Center. Linyi 276007, China
[2]  Shandong TongWei Information Engineering CO., LTD., Jinan 250000, China
[3]  School of Transportation, Southeast University, Nanjing 210096, China
*   Correspondence: sglkjk3@ly.shandong.cn (T.L.)

**Abstract:** Pavement cracks can result in the degradation of pavement performance. Due to the lack of timely inspection and reparation for the pavement cracks, with the development of cracks, the safety and service life of the pavement can be decreased. To curb the development of pavement cracks, detecting these cracks accurately plays an important role. In this paper, an automatic pavement crack detection method is proposed. For achieving real-time inspection, the YOLOV5 was selected as the base model. Due to the small size of the pavement cracks, the accuracy of most of the pavement crack deep learning-based methods cannot reach a high degree. To further improve the accuracy of those kind of methods, attention modules were employed. Based on the self-building datasets collected in Linyi city, the performance among various crack detection models was evaluated. The results showed that adding attention modules can effectively enhance the ability of crack detection. The precision of YOLOV5-CoordAtt reaches 95.27%. It was higher than other conventional and deep learning methods. According to the pictures of the results, the proposed methods can detect accurately under various situations.

**Keywords:** pavement crack detection; deep learning; attention mechanism

## 1. Introduction

Cracks on the pavement are a significant sign of potential damage and degradation in the performance and functionality of pavements. Generally, cracks of the pavement can be caused by heavy traffic, drastic temperature change, reflection from the base layers, and so on [1]. These cracks have a negative impact on the structure of the pavement, significantly decreasing the performance of the pavements. With cracks developing, the whole structure of the road can be influenced. The safety and service life of the road can be decreased to a certain degree. To curb the development of cracks, a frequent inspection is needed. By collecting various types of data for pavement conditions, the corresponding strategies and in-depth analysis can be made. According to the analysis results, timely and appropriate maintenance can be employed to repair the pavement and prevent its failure at an early stage of crack development. In this way, as for the pavement, the service life can be prolonged, and the performance and functionality can be maintained in a good condition.

As most previous studies have stated, collecting and analyzing the images of cracks is a primary way to realize the detection and classification of pavement cracks [2–4]. Generally, the pavement crack detection methods can be divided into two types based on whether the deep learning method is applied. They are the traditional methods and deep learning methods.

As for the traditional methods, visual inspection was a primary way to realize pavement crack detection in the early stages of the development of crack detection. Experienced workers need to inspect the whole road by walking. To accelerate the process of

visual inspection, the images of pavement cracks are collected by a slow-moving car. Then, researchers detect and classify the cracks by labor. The above methods need significant human intervention. Because of the extensive length of the road, these methods are time-consuming [5]. To improve the efficiency of the inspection for pavement cracks, some researchers employed image-based methods to detect cracks. Zhou et al. [6] applied the wavelet transform to pavement crack images. The images can be divided into different frequency sub-bands. The pavement cracks and background information can be divided into high- and low-amplitude wavelet coefficients. Similarly, Subirats et al. [7] developed a two-step pavement crack detection method by also applying the wavelet transform: firstly, establishing complex coefficient maps by a 2D continuous wavelet transform; then, employing the binary image of pavement cracks to indicate the presence of the cracks or not. In the study conducted by Li Peng et al. [8], a threshold-based detection method was developed to detect the cracks on an airport runway. This method can remove the road markings by segmentation and then the cracks can be detected. Besides that, Wei xu et al. [9] proposed an unsupervised method with the saliency and statistical characteristic of the pavement cracks. One of the novelties of this method is that the threshold was also applied in this paper to detect and classify pavement cracks. Besides that, some studies employed hand-crafted features to realize the detection and classification of pavement cracks [10–12]. The edge of the object can easily be acquired. Therefore, there was some research that utilized edge detection methods to recognize cracks [13].

With the development of computer science, employing deep learning methods to classify and detect pavement cracks has become emerging. In the study conducted by Zhun et al. [14], they utilized the convolutional neural network (CNN) to learn the structure of pavement cracks. Similarly, Fan et al. [15] proposed a novel network based on CNN to detect pavement cracks. This network combined the context information with low-level features for crack detection in a feature pyramid way. In addition, a new evaluation indicator was proposed based on the area of intersection. In sum, the imbalanced datasets were a usual problem in the above-mentioned deep learning methods. To solve this issue, Maeda et al. [16] applied the generative adversarial network (GAN) to generate more samples of pavement cracks. In this way, we have enough data to train the deep learning models. In addition, in this paper, to improve the reality of the pseudo cracks images, the Poisson blending was combined with GAN to generate the various pavement cracks. Furthermore, transfer learning, a segmentation method, and ensemble learning, etc., were applied in the detection and classification of pavement cracks [17–23]. However, these deep learning methods lack the ability to detect small objects. From the view of a car, the pavement cracks were usually small. In addition, the complicated lighting, weather conditions, etc., can have a negative impact in crack detection. Therefore, a more robust and accurate pavement crack detection method needs to be developed.

The main contributions of our work are summarized as follows:

(1) We propose a novel pavement crack detection method fusing with attention mechanisms to detect and classify the pavement cracks, which can obtain accurate detection results with only a small increase in computational load.

(2) A self-building pavement crack dataset collected in Linyi City has been proposed to be supplementary for the studies of cracks in terms of datasets.

(3) Experiments on the self-building pavement crack datasets and the public crack datasets [24] show that the proposed method significantly improves the accuracy of pavement crack detection over other methods.

The remainder of this article is organized as follows. Section 2 describes the establishment of the new datasets of pavement cracks and elaborates on the approach for detecting various types of cracks. The results and discussion are provided in Section 3. Finally, Section 4 delivers the conclusion of this work.

## 2. Materials and Methods

The proposed crack detection method was based on YOLOV5, and various attention mechanisms were added to it for enhancing the ability of crack detection. To achieve this goal, at first, we collected the data of various cracks at different sites to build the crack detection dataset. These cracks were then manually labeled to get the ground truths. For training and evaluating the proposed method, these data were divided into three types: training, validation, and test sets. Then, an improved YOLOV5 crack detection was proposed based on the self-building datasets.

Furthermore, the corresponding data processing, model training, and evaluation were mainly programmed by Python3.7. The specifications of the computer utilized for network training and evaluation are shown in Table 1.

**Table 1.** The specification of the computer used for model training and testing.

| Indicator | Value |
| --- | --- |
| GPU | Nvidia GeForce RTX 3070 |
| CPU | Intel Core i7-10700 |
| CUDA | 11.3 |
| CUDANN | 8.4.1 |
| Pytorch | 1.10.1 |

### 2.1. Dataset of Pavement Cracks

In this paper, we installed a high-resolution camera in a car to collect the data of various pavement cracks, as shown in Figure 1. Generally, the camera was installed in the middle of the front window, which can easily get a larger visual field.

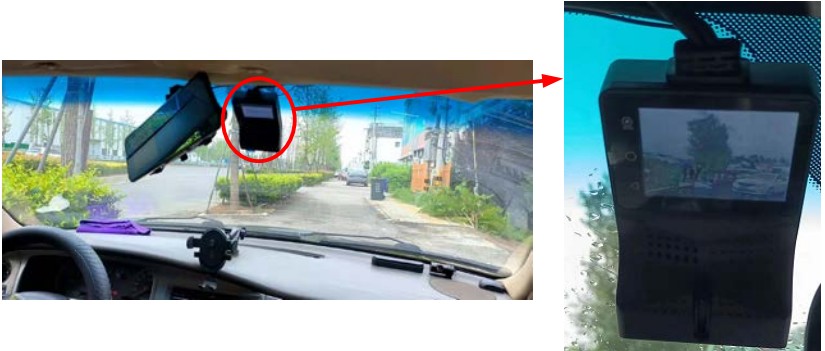

**Figure 1.** The high-resolution camera installed in a car to collect data of pavement cracks.

The specifications of the high-resolution camera are shown in Table 2.

**Table 2.** The specifications of the high-resolution camera.

| Indicator | Value |
| --- | --- |
| Image resolution | 1920×1080 |
| Scan FOV | 90° |
| Working temperature | –30 °C~60 °C |
| Working voltage | DC 9~18 V |
| Frequency | ≥10 Hz |

The images of the pavement cracks were collected from various roads, mainly in Linyi City, Shandong Province. The specific site was in Lanshan G205 and Hedong G205, as shown in Figure 2.

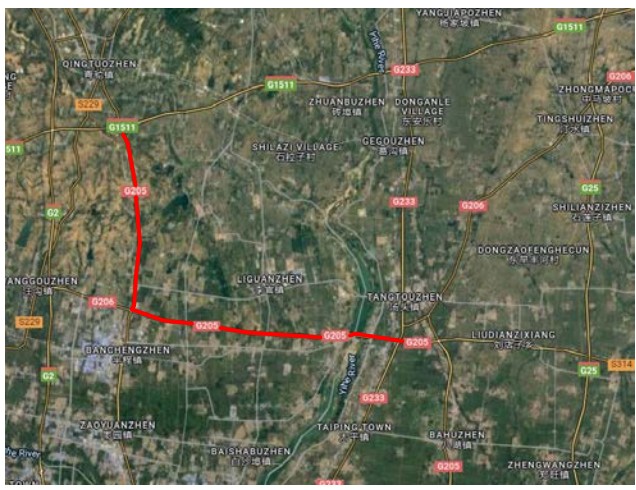

**Figure 2.** The locations for collecting pavement cracks.

To improve the robustness of the proposed method, the collected crack data consisted of different lighting conditions, including sunny and cloudy days. In addition, those pavement cracks with branches, leaves, water on the pavement, asphalt oil stains, and so on were also collected, resulting in noise in the images. In sum, 9650 images of pavement cracks were collected. In this paper, the ratio of training, validation, and testing sets were set as 6:2:2 [25]. Specifically, there were 5790 images of cracks for training, 1930 images of cracks for validation, and for testing the performance of the proposed crack detection method, respectively. To be noted, the ground truths of these collected pavement cracks were manually labeled through Roboflow. The bounding box was marked to detect cracks in images. Four types of cracks were classified based on the morphological features: longitudinal crack, transverse crack, alligator crack, and pothole, as shown in Figure 3. To be noted, the D00, D10, D20, and D40 were employed to represent the types of pavement cracks, respectively. The statistics of the pavement crack datasets are summarized in Table 3, as follows.

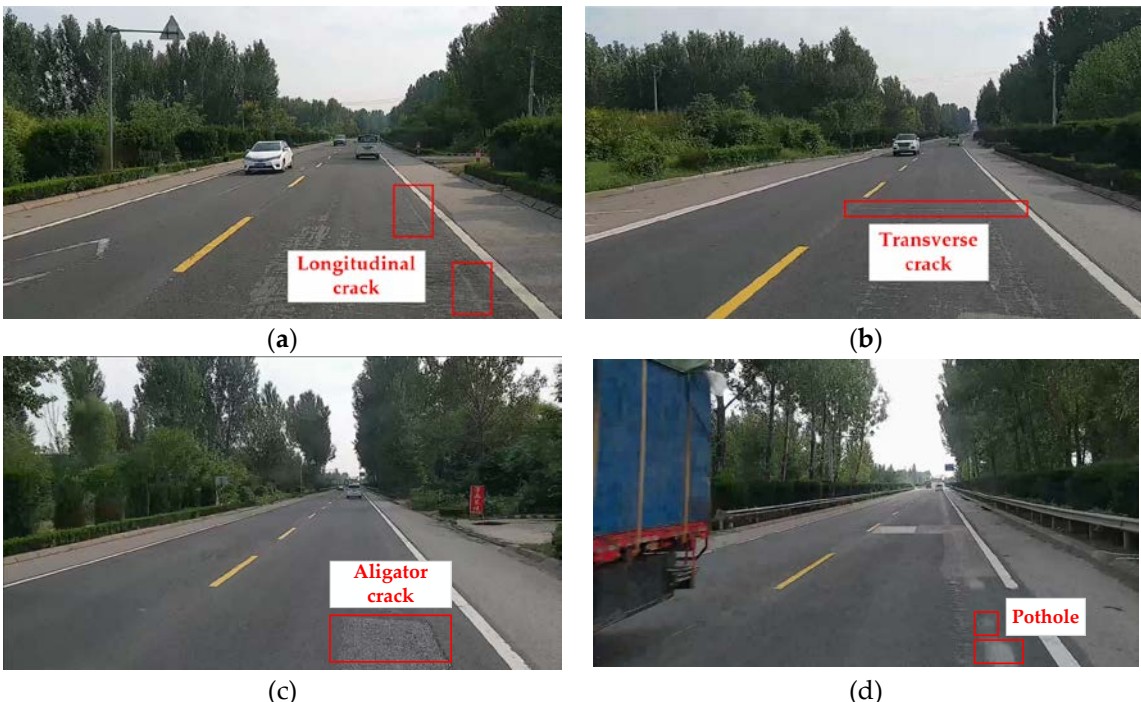

**Figure 3.** The types of pavement cracks: (**a**) longitudinal crack; (**b**) transverse crack; (**c**) alligator crack; (**d**) pothole.

**Table 3.** The statistics for the self-building pavement crack instances.

| Categories | Longitudinal Crack | Transverse Crack | Alligator Crack | Pothole |
|---|---|---|---|---|
| Number of instances | 4070 (42.17%) | 1202 (12.46%) | 3144 (32.58%) | 1234 (12.79%) |

*2.2. Architecture Design for Attention-Fused Crack Detection Network*

2.2.1. The Architecture of YOLOV5

You Only Look Once (YOLO) is an advanced, real-time object detection deep learning method, created by Redmore, Divvala, Girshick, and Farhail in 2016 [26], which belongs to R-CNN methods. Until now, various versions of YOLO have existed. In this paper, YOLOV5 was employed to detect and classify pavement cracks, which is the newest version of YOLO. In brief, YOLOV5 got a better performance based on the improvement in network architecture, data augmentation, and other aspects compared with YOLOV3 [27] and YOLOV4 [28].

In the design of network architecture, YOLOV5 can be divided into four parts: input module, backbone, neck network, and output module. Compared with the previous YOLOs, there were three parts improved in YOLOV5. (1) In the input module, the Mosaic [26], adaptive anchor box calculation, and adaptive image scaling were added to augment data. The principle of Mosaic is similar to CutMix [29], developed in 2019, to a certain degree. In CutMix, two pictures are jointed to enhance the input data. However, in Mosaic, four pictures are randomly selected to join together through random rotation, random scaling, and random distribution. In sum, there were three advantages to employing Mosaic. Firstly, the background of the detected object becomes rich. Secondly, a better model can be more easily trained with fewer GPU resources due to the combination of four pictures. Lastly, by applying random rotation, random scaling, and random distribution to generate the data, the number of small samples is increased. All of the advantages were beneficial to enhance the robustness of the crack detection model.

In the backbone, the revised CSPNet was employed to reduce the computational burden. Briefly, the original CSPNet mainly solves these three problems in the usual deep learning networks. (1) The gradient information was calculated repeatedly, resulting in the high computation load. CSPNet takes the gradient information into the feature maps all the time. In this way, the precision of the model can be retained, and the computation load can be reduced. (2) To enhance the efficiency of each neural unit in the network, CSPNet distributed the tasks to the layers of CNN equally to avoid extra consumption. (3) Generating the feature maps by crossing different channels can decrease the consumption of the memory effectively. Furthermore, there were types of revised CSPNet that were employed in YOLOV5. These were applied in the backbone (CSP1-X) and neck networks (CSP2-X), respectively. The structure of these CSPs is shown in Figure 4.

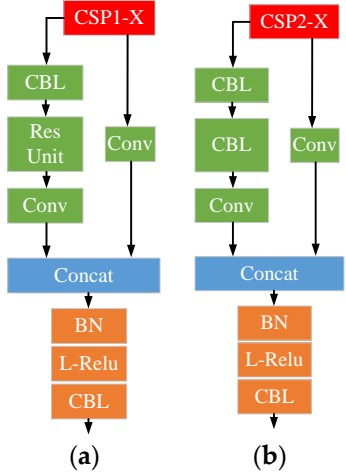

**Figure 4.** The architecture of CSPNets in YOLOV5: (**a**) CSP1-X; (**b**) CSP2-X.

In the output module, YOLOV5 employed a new loss function. Generally, the loss function in the object detection tasks was composed of two parts: classification loss and bounding box regression loss. As for the bounding box regression, the intersection over union (IoU) is an important indicator to present the difference between the detected bounding box and ground truths. The calculation can be denoted by Equation (1):

$$IoU = \frac{\left|A \cap B\right|}{\left|A \cup B\right|} \tag{1}$$

where $A$ and $B$ represent the area of the detected and true bounding box, respectively.

However, according to the definition of the IoU, it cannot fully reflect the degree of overlap between the detected and true bounding box when it is equal to 0. The gradient information cannot get propagated during the network training. Therefore, a new indicator GIoU (Generalized Intersection Over Union) [30] was introduced into YOLOV5 as the bounding box regression loss. The calculation of GIoU can be represented by Equation (2):

$$GIoU = IoU - \frac{\left|C/\left(A \cup B\right)\right|}{\left|C\right|} \tag{2}$$

where $C$ means the smallest enclosing convex object.

Therefore, the corresponding loss function can be denoted by Equation (3):

$$GIoU = 1 - GIoU \tag{3}$$

2.2.2. The Architecture of Attention Mechanisms

YOLO was not developed for civil engineering or road maintenance [20]. However, YOLOV5 has been tested to detect pavement cracks before. According to the detection results, it was found that the original YOLOV5 cannot get a satisfactory result. There existed some mistaken detection results. Therefore, to enhance the ability to recognize the critical features of cracks, the attention mechanism was added to YOLOV5.

In brief, the attention mechanism can be thought to be a method of acquiring a group of weight coefficients through network self-learning. To be noted, this method can pay attention to the critical area and abord the non-critical area in the images. That is to say the pavement cracks can get more attention compared with other background information, such as cars, branches, water, etc., by applying the attention mechanisms. Various attention modules have emerged over the past years. In this paper, four types of attention modules were added to YOLOV5 for improving the performance of pavement crack detection. They were SENet [31], ECANet [32], CBAM [33], and CoordAtt [34].

SENet, revised on ResNet, can adjust the attention to different channels according to the importance of channels. ECANet (Efficient Channel Attention) got an improvement based on SENet. To avoid generating redundant parameters, the ECA replaced the fully-connected layers with the one-dimensional convolutional layer. In this way, the ECA was more efficient than SENet. In conclusion, the SENet and ECA both focused on the information from various channels. The spatial information was neglected by these attention modules. Therefore, to enhance the ability to extract spatial features, CBAM was developed. This attention module can not only pay attention to the channel information, but also spatial information. Furthermore, according to some previous studies, the location information in channels plays an important role in attention map generation. By adding the spatial coordinate into the feature maps, the CoordAtt can make the light networks pay attention to a larger area with less computation. The structure of these four attention modules is shown in Figure 5.

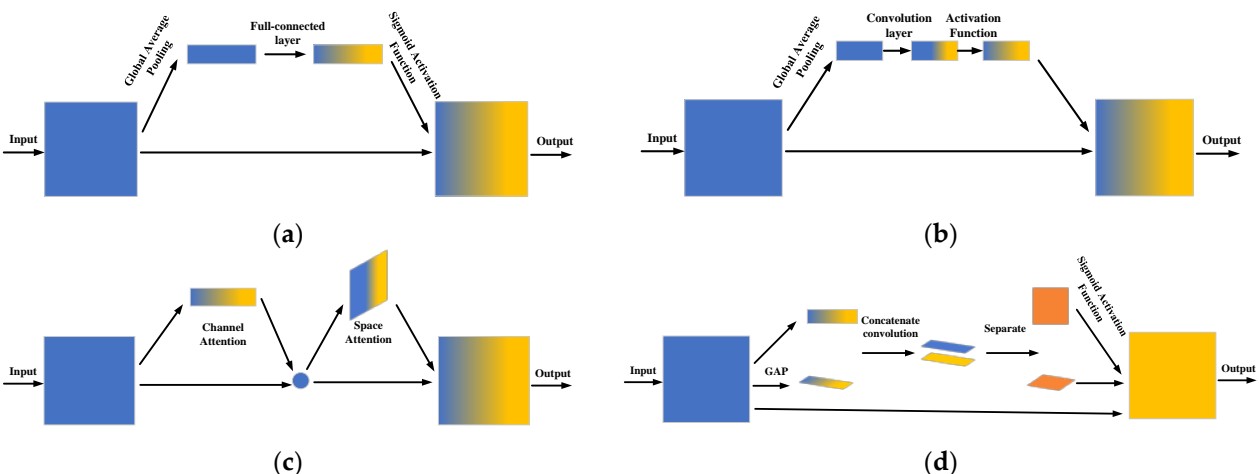

**Figure 5.** The structure of attention modules: (**a**) SENet; (**b**) ECANet; (**c**) CBAM; (**d**) CoordAtt.

### 2.2.3. Improved Crack Detection with Fusing Attention Mechanism

As previous studies have stated, increasing the depth of the network can improve the performance of object detection [35]. The backbone in YOLOV5 plays a critical role in extracting crack features compared with other parts. Therefore, in this paper, various attention modules were added to the backbone, rather than replacing some modules with them. The architecture of the original and improved backbones is shown in Figure 6.

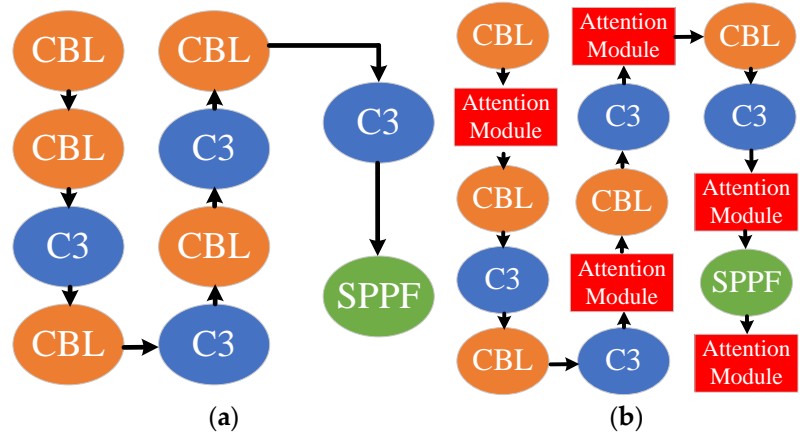

**Figure 6.** The architecture of backbone in YOLOV5: (**a**) the original backbone; (**b**) the improved backbone with attention modules.

As shown in Figure 6a, the backbone of the original YOLOV5s was composed of CBL, C3, and SPPF. In this paper, CBL is a kind of module composed of a Convolution layer, Batch normalization, and Leaky relu activation function. C3 consists of three convolution layers and multiple ResNet units. As for SPPF, it is a special pooling module including three types of max pooling. From Figure 6b, it can be found that the improved YOLOV5s backbone has several added attention modules to further enhance the ability of feature extraction. There were five locations for adding these attention modules. The reason for the first location was that the attention modules located in these two sites can sufficiently extract features of pavement cracks compared with the original YOLOV5. As for the second and third locations, the features generated from these two locations would be sent to the detection head. After fusing these features, the detection head can utilize them to predict the bounding boxes for cracks, which mostly determine the detection performance. The fourth and fifth locations were circled with SPPF. This module can easily transform

random-size feature maps into fixed-size feature maps. These two attention modules can further retain and extract the critical features of cracks.

### 2.3. Training Procedure

To get a more precise pavement crack detection model, the proposed method has been trained with 500 epochs, and the batch size and initial learning rate were set to be 64 and 0.01, respectively. The training time can generally reach 7 h on average.

## 3. Results and Evaluation

### 3.1. Evaluation Metrics

To sufficiently evaluate the performance of the proposed method, three evaluation indicators were applied. They were Recall, Precision, and mAP. Generally, the detected objects can be classified into four types based on the relationship between the predicted and true results, as shown in Table 3.

TP (true positives) and TN (true negatives) mean the predicted result was the same as the true result. FN (false negatives) and FP (false positives) mean the predicted result was different from the true result. In terms of pavement cracks, the TP and TN were the correctly detected pavement cracks and non-cracks. The FP was the non-cracks wrongly detected as cracks. Correspondingly, the FN was the cracks wrongly detected as non-cracks. Based on Table 4, the definition of these three evaluation indicators can be acquired.

**Table 4.** The types of detected pavement cracks.

| Ground Truths | Predicted | |
|---|---|---|
| | True Detection Result | False Detection Result |
| True detection result | TP | FN |
| False detection result | FP | TN |

As shown in Equation (4), the calculation of the precision denotes the percentage of the true positives amongst the predicted result consisting of true positives and false positives. The recall represents the percentage of true positives amongst the predicted result composed of true positives and false positives, as shown in Equation (5). Based on the precision and recall, the F1-score can be calculated as shown in Equation (6):

$$\text{Pr}\,ecision = \frac{TP}{TP + FP} \tag{4}$$

$$\text{Re}\,call = \frac{TP}{TP + FN} \tag{5}$$

$$F1 = 2 \times \frac{\text{Pr}\,ecision \times \text{Re}\,call}{\text{Pr}\,ecision + \text{Re}\,call} \tag{6}$$

Furthermore, in this paper, the average precision, AP, was also employed to evaluate the performance of the developed method. This indicator can be obtained by calculating the area enclosed by a curve, whose x-axis and y-axis were recall and precision, respectively. The mAP, mean average precision, means the average of the AP for all categories.

### 3.2. Pavement Crack Detection Results

Generally, the networks can get better performance with more training epochs. However, this way can also result in the overfitting issue. In this paper, to avoid this issue, the coefficient weights were selected based on this principle: in the validation sets, the weights were selected when the model got the best performance, rather than the weights being selected in the last epoch. Then, employing the weight coefficients to detect the pavement cracks in the test sets, various indicators were applied to evaluate the performance among different pavement crack detection models.

The performance evaluation results are shown in Table 5. According to Precision, adding attention modules can effectively improve the performance of detecting pavement cracks. Specifically, adding CoordAtt can obtain the best performance among these five models. Compared with the original model, the precision was higher by 4.36%. In addition, the performance of YOLOV5s-CoordAtt was also superior to YOLOV3 [27].

**Table 5.** The evaluation results of various pavement crack detection methods (self-building datasets).

| Methods | Precision | Recall | F1 | Map |
|---|---|---|---|---|
| YOLOV5s | 90.91% | 85.59% | 88.17% | 90.71% |
| YOLOV5s-SE | 93.21% | 82.41% | 87.47% | 90.57% |
| YOLOV5s-ECA | 94.76% | 83.21% | 88.61% | 91.07% |
| YOLOV5s-CBAM | 94.58% | 80.60% | 87.03% | 89.40% |
| YOLOV5s-CoordAtt | 95.27% | 83.45% | 88.96% | 91.81% |

As shown in Table 5, various attention mechanisms have different impacts on pavement crack detection performance. In conclusion, the attention modules can enhance the crack detection performance. From the precision, it can be found that YOLOV5s-ECA performed better than YOLOV5s-SE. The reason for this result was that the ECANet replaced the full-connected layer with a convolution layer, reducing the computation load and improving the performance. Compared with YOLOV5s-CBAM, YOLOV5s-SE had a poor performance. This was because YOLOV5s-CBAM put the attention on the channel and spatial information at the same time. A pavement crack is a kind of object with irregular shapes. Attaching attention on its spatial information further contributed to extracting crack features. As for YOLOV5s-CoordAtt, this attention module added the location information into the channel attention information; this method can effectively improve the performance of pavement crack detection.

The test results are shown in Figure 7, and the proposed method can accurately classify and detect these four types of pavement cracks in the images. In addition, from Figure 7d, it can be found that the proposed model can also recognize pavement cracks under poor light conditions, such as on cloudy days.

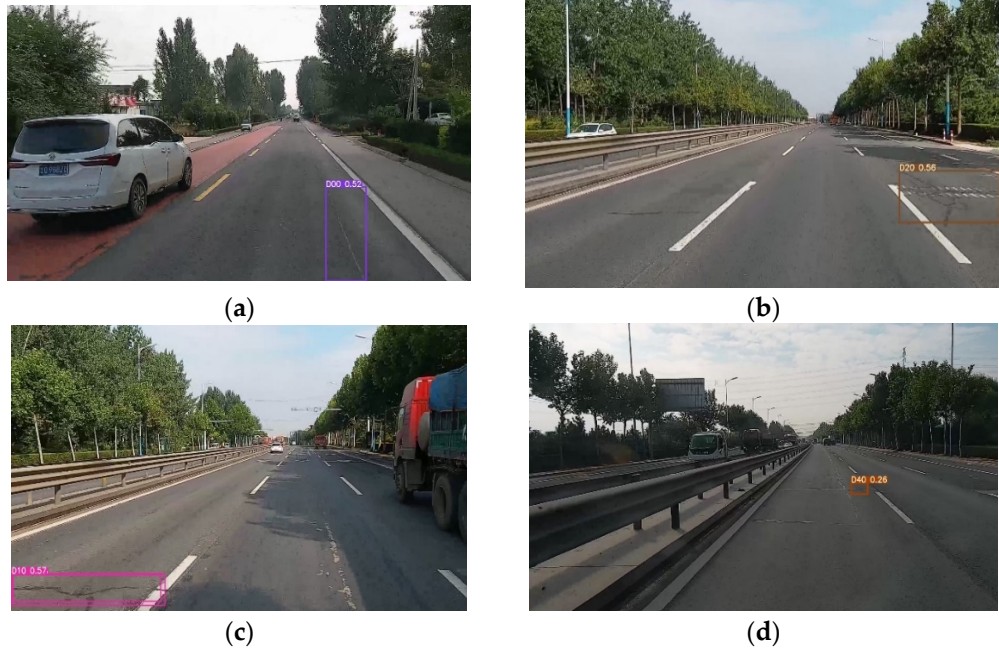

**Figure 7.** The pavement crack detection results of YOLOV5s-CoordAtt: (**a**) longitudinal crack; (**b**) transverse crack; (**c**) alligator crack; (**d**) pothole.

From Figure 7, it can be found that the proposed methods can detect these four types of pavement cracks accurately. However, there still existed some situations where the detection results showed that the categories of pavement cracks were confused by the proposed methods. As shown in Figure 8a, the longitudinal cracks were recognized as potholes by mistake. Moreover, the longitudinal cracks can also be recognized as transverse cracks due to the shadow, as shown in Figure 8b. These results showed that the proposed pavement crack detection method confused the longitudinal cracks with other kinds of cracks in some circumstances, due to the similarity of clustered longitudinal cracks and potholes or transverse cracks. Additionally, other noises in images, including lighting conditions, shadows, and pavement markings, make it more difficult for the proposed approach to recognize the types of pavement cracks.

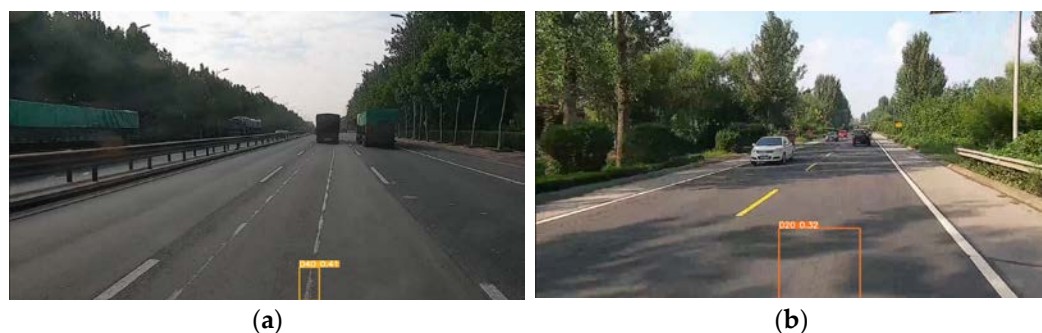

(**a**)　　　　　　　　　　　　　　　　　　　　　　　　　　　　　　(**b**)

**Figure 8.** False detection results of pavement cracks: (**a**) longitudinal crack was mistakenly detected as pothole; (**b**) longitudinal crack was mistakenly detected as alligator crack.

Furthermore, to sufficiently test the performance of the proposed methods, the public dataset [24] was also employed to evaluate the performance of the proposed methods and other existing approaches [36–38]. The corresponding results are shown in Table 6.

**Table 6.** The evaluation results of various pavement crack detection methods (public datasets).

| Methods | Precision | Recall | F1 | Map-0.5 |
|---|---|---|---|---|
| YOLOV5s | 74.9% | 61.4% | 67.4% | 67.1% |
| YOLOV5s-SE | 76.9% | 49% | 59.9% | 68.4% |
| YOLOV5s-ECA | 74.7% | 52.1% | 61.4% | 59.9% |
| YOLOV5s-CBAM | 72.8% | 52.3% | 60.8% | 68.7% |
| YOLOV5s-CoordAtt | 77.8% | 49.5% | 60.5% | 71.4% |
| YOLOV3 [38] | 76.5% | 65.2% | 70.4% | 70.3% |
| FasteRCNN [36] | 74.2% | 51.7% | 60.9% | 59.2% |
| YOLOV4[37] | 71.5% | 66.2% | 68.7% | 52.7% |

As shown in Table 6, YOLOV5s fused with the Coordinate Attention module reached the highest performance compared to other methods in the public datasets. As shown in Table 6, it can be found that the precision of FaseRCNN, YOLOV4, YOLOV3, YOLOV5, and YOLOV5s fusing with CoordAtt were 74.2%, 71.5%, 76.5%, 74.9%, and 77.8%, respectively. Besides that, the mAP of these five models were 59.2%, 52.7%, 70.3%, 67.1%, and 71.4%, respectively. In conclusion, the proposed method, YOLOV5s fusing with CoordAtt, can get the best performance compared to other existing approaches. In conclusion, from Table 7, it can be seen that the proposed crack detection method (YOLOV5s fusing with CoodAtt) can get better performance than the existing models (the original YOLOV5s, YOLOV3, YOLOV4, and fasteRCNN). Besides that, we have also made a comparison with the self-building datasets between the proposed methods and the existing approaches

shown in Table 8. It can be found that the proposed method can also get the highest performance among various recent crack-detection-related approaches in the self-building datasets.

**Table 7.** The comparison between ours and existing models (public datasets).

| Methods | Precision | Recall | F1 | Map |
|---|---|---|---|---|
| YOLOV4[37] | 71.5% | 66.2% | 68.7% | 52.7% |
| FasteRCNN [36] | 74.2% | 51.7% | 60.9% | 59.2% |
| YOLOV3[38] | 76.5% | 65.2% | 70.4% | 70.3% |
| YOLOV5s | 74.9% | 61.4% | 67.4% | 67.1% |
| Ours | **77.8%** | 49.5% | 60.5% | **71.4%** |

**Table 8.** The comparison between ours and existing models (self-building datasets).

| Methods | Precision | Recall | F1 | Map |
|---|---|---|---|---|
| YOLOV4[37] | 87.6% | 81.8% | 84.6% | 79.7% |
| FasteRCNN [36] | 86.9% | 80.7% | 83.6% | 81.4% |
| YOLOV3[38] | 88.4% | 78.3% | 83.0% | 82.5% |
| YOLOV5s | 90.9% | 85.6% | 88.1% | 90.7% |
| Ours | **95.3%** | 83.4% | 88.9% | **91.8%** |

## 4. Conclusions

Automatic pavement crack detection has become more and more important in civil engineering. This is because accurate and real-time road inspections are essential for road maintenance. However, the traditional object detection model was not designed to detect and classify pavement cracks. Compared to those usual objects, the pavement cracks are smaller. Therefore, a pavement crack detection model with attention modules was developed to improve the performance of classifying and detecting pavement cracks. The main novelty of the proposed method is that various attention modules were introduced into the original networks to improve the ability of extracting the features of pavement cracks. Moreover, a dataset of pavement cracks collected in Linyi City, Shandong Province was developed for further research in pavement crack detection. From the test results, it was found that adding attention modules can effectively improve the performance of the pavement cracks. The precision of the proposed method was 4.36% higher than the original model. Furthermore, according to the detection results evaluated in the public datasets, the proposed methods can also obtain a better performance than other existing models. However, some limitations in this paper still exist. The confidence scores of pavement cracks are not so high. Compared with previous studies, the installment may play an important role in this issue. A better installment position of the high-resolution camera will be studied in the future. The classes in the developed datasets need to be supplemented in the future. The bounding box may not describe the shape of pavement cracks so accurately. The combination of the segmentation and detection of pavement cracks will also be studied in the future.

**Author Contributions:** Conceptualization, J.R. and G.Z.; methodology, J.R.; validation, Y.M., D.Z., and T.L.; formal analysis, J.R.; investigation, G.Z.; resources, T.L.; data curation, D.Z.; writing—original draft preparation, J.R.; writing—review and editing, D.Z.; visualization, Y.M., and J.Y.; supervision, T.L.; project administration, T.L.; funding acquisition, D.Z. All authors have read and agreed to the published version of the manuscript.

**Funding:** This research was funded by Research and Application of Intelligent Monitoring Technology for Highways based on Computer Vision, grant number 2022B53-02.

**Institutional Review Board Statement:** Not applicable.

**Informed Consent Statement:** Not applicable.

**Data Availability Statement:** Not applicable.

**Conflicts of Interest:** The authors declare no conflict of interest.

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
