# Peer review of "Automatic Pavement Crack Detection Fusing Attention Mechanism"

_electronics, doi:10.3390/electronics11213622_

Round 1

Reviewer 1 Report

The contributions of the article are not clear to me. Please clearly present the main contributions in the Introduction, e.g., improved YOLO, self-built dataset.

The discussions of relevant works are very limited. I will suggest to add a new section to discuss the related fields, such as pavement crack detection as well as neural attention mechanisms.

In addition, some existing works that are based on neural attention should be included, like Motion-attentive transition for zero-shot video object segmentation, Cascaded human-object interaction recognition.

It will be better to summarize the statistics of the built dataset within a table.

The results only on self-built datasets are not sufficient. I will suggest to test the proposed model on existing public datasets.

From Table 4, we see that different attention modules have different impacts to performance. Some insights should be given regarding these results.

Author Response

The corresponding response has been attached in the file Reply_to_Reviewer_1.docx.

Reviewer 2 Report

In this paper, the authors present a pavement detection model. The work is good but still, there is some scope for improvement required:

1.     In the methodology section, the authors mentioned that they created their own dataset, but the proposed work should be compared with an existing global dataset for better understanding.

2.     In figure 6, the authors use an acronym as CBL... What is it??  The difference between original and improvised architecture should be clearly mentioned in the manuscript.

3.     As the Authors claim that their work is novel, then it should be compared with an existing model which not strongly discussed.

4.     In equation 6, the authors show the f1 score calculation equation, but the value of the f1 score is not being calculated. Why?

5.     Is the mAP (line no 247) and Map-0.5 (Line no. 262) are same? If yes then why are two different acronyms used?

6. On page no. 7, figure 5, represents “The structure of attention modules” whereas on page no. again figure 5 represents “The pavement cracks detection results of YoloV5s-CoordAtt”. Why this very casual approach to writing?

7.     There few works can be found on pothole detection authors may cite the below paper to strengthen their reference articles:

1.       Choudhury, A., Ramchandani, R., Shamoon, M., Khare, A., Kaushik, K. (2020). An Efficient Algorithm for Detecting and Measure the Properties of Pothole. In: Mandal, J., Bhattacharya, D. (eds) Emerging Technology in Modelling and Graphics. Advances in Intelligent Systems and Computing, vol 937. Springer, Singapore. https://doi.org/10.1007/978-981-13-7403-6_40

2.       Ahmed, K.R. Smart Pothole Detection Using Deep Learning Based on Dilated Convolution. Sensors 202121, 8406. https://doi.org/10.3390/s21248406

Author Response

The corresponding response has been attached in the file Reply_to_Reviewer_2.docx.

Round 2

Reviewer 2 Report

The revised version looks good and can be accepted. 

Comments.

On page no. 7 and 9, both figures mentioned as figure no 5. I think two different figures can not be having the same number.

Author Response

Point 1.On page no. 7 and 9, both figures mentioned as figure no 5. I think two different figures can not be having the same number.

Response 1: Thanks for pointing out this point. We have checked the whole paper carefully. The corresponding problems have been corrected.